# REAL-TIME ROUTING UNDER PARTIAL OBSERVABILITY: INFORMATION-EFFICIENT POLICIES FOR CONNECTED VEHICLES

## ABSTRACT

Real-time navigation in urban road networks requires making sequential routing decisions with incomplete and noisy information. Recent advances in IoT infrastructure and vehicle-to-everything (V2X) technologies enable connected vehicles to communicate with roadside units and traffic signals in real time. However, in practice, communication bandwidth and deployment budgets severely restrict the number of intersections that can be queried at each decision step, creating a partially observable environment for real-time navigation. Existing pipelines which separately train predictors of traffic states and then apply non-differentiable routing solvers struggle under such conditions, as they assume access to dense and complete sensing. In this paper, we present an end-to-end differentiable framework that jointly addresses vehicle-to-infrastructure(V2I) information acquisition, traffic state inference, and dynamic routing optimization. In the proposed framework, a learnable selection module proactively determines which intersections to query under communication constraints, followed by a spatio-temporal aware encoder that infers network-wide travel costs from the resulting sparse signals, and a differentiable soft shortest-path decision decoder computes re-routing strategies while allowing gradients of downstream travel cost to flow back through the entire pipeline. This tight coupling aligns model training with the true system objective of minimizing vehicle travel time. Experiments on microscopic simulation with city-scale networks demonstrate that our approach outperforms comparable baselines in travel efficiency while requiring only minimal communication. By integrating selective information acquisition and differentiable decision-making, our framework advances real-time urban navigation under partial observability and provides a scalable path toward deployment in intelligent transportation systems.

## 1 INTRODUCTION

Real-time navigation at urban intersections is a cornerstone of intelligent transportation systems, enabling vehicles to traverse complex road networks with minimal delay and improved safety. With the rapid deployment of connected vehicles and roadside units, both infrastructure and vehicles are increasingly capable of transmitting rich information about traffic states and driving intentions.

**Challenge.** A fundamental obstacle remains: *at current stage, only partial information can be made available at every decision step*. Due to budget, bandwidth, and computation constraints, the system can only query a limited subset of vehicles or roadside sensors at a time (Fabris et al., 2025; Almutairi & Owais, 2025). This limitation creates a partially observable environment where navigation decisions must be made with incomplete knowledge.

**Limitations of existing methods.** The dominant paradigm in the described real-world decision-making system is *predict-then-optimize*: a predictive model is first trained to estimate traffic states—such as link travel times, densities, or queue lengths—from historical and current measurements, and the resulting estimates are then passed to an optimization module such as a shortest-path solver or dynamic traffic assignment (Tian et al., 2023; Jiang et al., 2025; Huang et al., 2025). While this pipeline has been widely adopted, it critically relies on the assumption that the traffic state is either fully observable or can be reconstructed from dense sensor coverage. For example, most

spatio-temporal graph models for traffic prediction assume access to complete network-level speed and flow measurements (Li et al., 2018b; Yu et al., 2018; Guo et al., 2019; Wu et al., 2019; Zhao et al., 2019; Zheng et al., 2020; Cao et al., 2020), and many routing studies treat link travel times as exogenous inputs (Sun et al., 2021; Liu & Han, 2020; Bast et al., 2016b; Levering et al., 2022). In practice, however, connected and automated vehicle (CAV) networks with vehicle-to-everything (V2X) communication operate under stringent bandwidth, infrastructure budget, and penetration constraints. As a result, only a subset of vehicles or roadside sensors can actively transmit information at each decision step, leading inevitably to partial observability of the network state (Campolo et al., 2017; Feng et al., 2021; Li et al., 2021; Sun & Ban, 2023). Under such conditions, conventional predict-then-optimize pipelines may yield suboptimal routing recommendations.

**Our approach.** To deal with this challenge, we argue that *partial information acquisition, global travel cost inference, and navigation optimization must be addressed jointly rather than as isolated stages*. We propose an end-to-end framework that integrates:

- **Proactive Information Acquisition.** A learnable vehicle-to-infrastructure(V2I) acquisition module that proactively determines which roadside sensors to query under strict communication and budgetary constraints, realized via a principled differentiable discrete selection network.
- **Full Travel Cost Inference.** A spatio-temporal aware module that infers global traffic states from partial and noisy observations acquired, exploiting both road network topology and temporal traffic dynamics to infer edge travel costs.
- **Differentiable Routing Head.** A soft Bellman-based decision layer that transforms the non-differentiable shortest-path optimization into a smooth surrogate, producing differentiable estimates of cost-to-go and routing distributions. This allows downstream travel-time objectives to directly supervise upstream state inference and information acquisition in an end-to-end manner.

This joint formulation departs from conventional predict-then-optimize pipelines by explicitly coupling sensing, prediction, and control within a single learning objective. In doing so, it aligns model training directly with the ultimate system goal: minimizing travel time and delay. Through extensive microscopic simulation on real-world road networks, we show that our framework consistently outperforms comparative baselines while achieving near-oracle efficiency with minimal communication cost. Moreover, the differentiable optimization layer enables millisecond-level decision updates and inference, making the approach feasible for real-world deployment at scale.

**Contributions.** In summary, our contributions are threefold:

1. We formalize real-time intersection navigation under communication-constrained partial observability, highlighting proactive information acquisition as a core component of the re-routing decision problem.
2. We propose an end-to-end differentiable framework that unifies selective sensing, full state reconstruction, and learnable navigation optimization.
3. We empirically validate the framework across diverse traffic scenarios, demonstrating performance in travel efficiency. Codes will be made public to facilitate research in this area.

## 2 RELATED WORKS

### 2.1 DYNAMIC VEHICLE NAVIGATION

Dynamic vehicle navigation can be naturally framed as a dynamic shortest path (DSP) problem, where edge costs evolve over time or are updated in response to real-world events such as congestion, incidents, or environmental conditions (Baum et al., 2015; Meuser et al., 2019). Unlike static shortest path formulations (Jiang et al., 2024), where link weights are fixed and deterministic, dynamic navigation must account for time-varying or stochastic costs and adapt to incomplete observations (Madkour et al., 2017; Ríos et al., 2021). A common specialization is the time-dependent shortest path (TDSP) problem, in which travel times are modeled as functions of the departure time, typically under the first-in-first-out assumption. In contrast, general DSP formulations capture

abrupt or unpredictable disruptions that require continuous re-optimization. Classical approaches extend Dijkstra- or A*-based algorithms to time-dependent networks (Dijkstra, 1959; Hart et al., 1968), and include techniques such as dynamic discretization discovery and formulations of the time-dependent traveling salesman problem (TDTSP) (Hansknecht et al., 2021; He et al., 2022). While these methods can provide optimality guarantees under specific assumptions, they face challenges in terms of scalability and efficiency when applied to large urban road networks subject to uncertainty. To address these limitations, the literature has advanced toward online and stochastic variants, including dynamic vehicle routing problems (Ríos et al., 2021; Adamo et al., 2024), robust shortest paths (Filippi et al., 2025), and stochastic travel-time models (Sever et al., 2018).

In parallel, learning-based approaches have emerged as a promising paradigm. Reinforcement learning methods treat navigation as a sequential control task, enabling vehicles to adaptively react to evolving traffic states (Koh et al., 2020; Li et al., 2024; Sun et al., 2023). Graph neural networks have been leveraged to approximate routing policies or shortest-path computations directly on large-scale networks, offering scalability and generalization across unseen topologies (Liu & Meidani, 2025; Chen et al., 2024). Furthermore, A complementary perspective comes from frameworks that explicitly integrate learning-based prediction with optimization, which explicitly couple predictive modeling with downstream decision-making (Vanderschueren et al., 2022; Shah et al., 2022).

## 2.2 Predict-Then-Optimize Paradigm

Recent advances in machine learning for decision-making, especially in transportation, logistics, and energy, largely adopt a *predict-then-optimize* (PtO) paradigm: learning predictive models for costs or states and applying these predictions as inputs to routing, scheduling, or control optimization. In vehicle navigation problems, PtO methods are widely adopted, where predicted traffic states feed into shortest path solvers, dynamic traffic assignment (DTA) models, or signal control strategies (Tian et al., 2023; Jiang et al., 2025; Huang et al., 2025). While effective with dense sensing, this pipeline critically relies on the assumption that traffic states are fully observed, and may degrade under partial observability. More importantly, although this decoupled prediction and optimization simplifies system design, it introduces a mismatch between predictive objectives (e.g., minimizing prediction MSE) and optimization objectives (e.g., minimizing travel time). Recent decision-focused learning methods (Vanderschueren et al., 2022; Shah et al., 2022) address this limitation by aligning model training directly with downstream decision quality, offering a more principled integration of learning and optimization. However, the effect of selectively leveraging partial information within this paradigm, as well as its impact on downstream routing performance in urban networks, have not been systematically studied.

## 2.3 Traffic State Estimation under Partial Observability

Traffic state estimation (TSE) aims to infer spatio-temporal traffic conditions—such as speeds, flows, densities, or queues—from sparse, noisy, and often heterogeneous observations (Seo et al., 2017; Wang et al., 2025). Recent advances in physics-informed deep learning (PIDL) embed conservation laws into neural architectures, improving estimation quality under limited sensing and enhancing physical consistency (Shi et al., 2021; Di et al., 2023). At the intersection level, connected vehicle (CV) trajectories have been leveraged to estimate volumes and queues even under low penetration rates (Zheng & Liu, 2017; Jia et al., 2025), while uncertainty-aware models provide confidence bounds for such inferences (Sun & Ban, 2023; Wang et al., 2024). These developments highlight the progress in robust state inference under partial observability, yet most approaches still assume a fixed observation set and stop short of linking estimation with downstream decision-making. These advances underscore the importance of robust inference under partial observability. However, existing methods are largely designed for accurate reconstruction or uncertainty quantification, and the observation set itself is typically assumed fixed. In contrast, the effect of selectively acquiring partial observations on downstream decision quality has received little attention.

# 3 Preliminaries

We model the urban transportation network as a directed graph $G = (V, E)$, where each node $v \in V$ denotes an intersection and each edge $e = (u, v) \in E$ represents a directed road segment. We en-

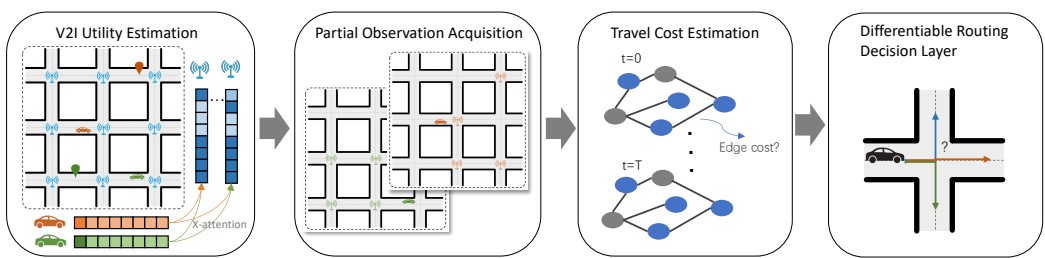

Figure 1: Overall framework of our proposed method.

code the structural layout of the road network through its adjacency matrix $A \in \{0,1\}^{|V| \times |V|}$, where $A_{ij} = 1$ if intersection $i$ and $j$ are directly connected by an edge. Each edge is associated with a time-dependent travel cost $w_e(t)$ that reflects dynamic traffic conditions such as congestion, incidents, or control signals, but are not directly available. At time $t$, a vehicle seeks a path $\pi = (e_1, \ldots, e_k)$ from a source edge to a destination edge. The total travel time of $\pi$ is $C(\pi, t) = \sum_{i=1}^{k} w_{e_i}(t_i)$, where $t_i$ is the entry time into edge $e_i$. The *dynamic re-routing problem* is to adaptively update $\pi$ as traffic conditions evolve, in order to minimize $C(\pi, t)$.

In our setup, dynamic re-routing is enabled by V2I communication: vehicles can exchange information with traffic signals at selected intersections to obtain local traffic states. Due to practical limitations such as bandwidth and communication constraints, a vehicle cannot query the entire network at each step. Instead, at each decision step, the vehicle selects a subset of $k$ intersections with which to communicate, under communication constraints. Formally, let $S_t \subseteq V$ with $|S_t| = k$ denote the selected intersections at time $t$. Through V2I interactions with each $s \in S_t$, the vehicle receives partial observations $\mathcal{O}_t$ of local conditions, such as signal phases, queue lengths, and local vehicle trajectories. This observation acquisition step determines which parts of the network are visible to the vehicle, while the rest remain hidden.

Based on the partial observation $\mathcal{O}_t$ and its current position $v_t$, as well as its destination goal, the vehicle then makes a re-routing decision $a_t \in \mathcal{A}(v_t)$, where $\mathcal{A}(v_t)$ is the set of feasible outgoing edges. The decision-making policy thus involves two stages at each step: (1) choosing informative intersections to query under the top-$k$ constraint, and (2) selecting the next edge to traverse so as to avoid congestion and minimize the overall travel cost. This setup couples information acquisition with dynamic routing, highlighting the importance of observation selection in achieving efficient navigation under partial observability.

## 4 METHODOLOGY

We design an end-to-end framework for dynamic re-routing under partial observability in connected intersections. Our method, as shown in Figure 1, integrates three critical components: (1) learning V2I Informativeness under strict constraints, (2) inferring travel costs from partial observability, (3) computing routes with a soft differentiable Bellman layer. This section describes each part in detail.

### 4.1 LEARNING V2I INFORMATIVENESS UNDER CONSTRAINTS

**Vehicle context.** The state of the querying vehicle is represented by a set of categorical embeddings that capture both its instantaneous position and its navigation objective. Specifically, we encode (i) its current intersection $v_t$, (ii) the incoming approach direction at $v_t$, (iii) the destination intersection $d$, and (iv) the approach direction of the destination edge. These one-hot representations, when combined, provide a compact but expressive characterization of the vehicle's situational context, aligning the scoring process with its intended routing goal.

**Intersection selection.** Selecting $k$ intersections for V2I communication under budget constraints is inherently a combinatorial optimization problem:

$$z^* = \arg\max_{z \in \mathcal{C}_k}\langle\theta, z\rangle, \quad \mathcal{C}_k = \{z \in \{0,1\}^{|V|} \mid \sum_{i=1}^{|V|} z_i = k\}, \tag{1}$$

where $z$ is a binary indicator vector encoding the queried intersections. The feasible set $\mathcal{C}_k$ grows combinatorially with $|V|$, and the argmax operator is discrete and non-differentiable. This breaks the gradient flow required for end-to-end training, and naïve relaxations (e.g., soft top-$k$ (Jang et al., 2017; Maddison et al., 2017; Kool et al., 2019)) typically fail to preserve the combinatorial structure and yield poor alignment with routing objectives.

To address this challenge, we frame intersection selection as sampling from a structured exponential family distribution over $\mathcal{C}_k$, parameterized by the utility scores $\theta$. This probabilistic perspective transforms the hard argmax into a stochastic decision process while strictly preserving the top-$k$ cardinality constraint. Inspired by recent work on implicit likelihood matching for discrete distributions (Niepert et al., 2021), we adapt the underlying principle to our setting: rather than optimizing for generic sampling fidelity, we construct a surrogate *loss-informed target distribution* $q(z; \theta')$ that places more probability mass on intersection subsets leading to lower routing cost, and update the model distribution $p(z; \theta)$ accordingly. Practically, this is realized through perturb-and-MAP sampling, where structured noise is added to $\theta$, a top-$k$ MAP selection is performed, and a second target sample is obtained after shifting $\theta$ toward the task-loss gradient. The gradient estimator, given by the difference between these two MAP solutions,

$$\nabla_\theta \mathcal{L} \approx \text{MAP}(\theta + \epsilon) - \text{MAP}(\theta' + \epsilon), \tag{2}$$

provides a low-variance update direction that allows end-to-end optimization of the utility scoring network. In this way, the discrete intersection selection policy is directly trained with respect to its impact on downstream navigation performance. This design has two key advantages: (i) it respects the combinatorial nature of the top-$k$ constraint without resorting to crude continuous relaxations, and (ii) it directly aligns the intersection query policy with downstream routing regret, ensuring that communication choices are optimized not for generic information gain but for their impact on decision quality.

**Context-Aware Utility Estimation.** Given the problem formulated as above, we compute utilities via vehicle-intersection cross-attention so that the vehicle context selectively attends to informative intersections. Let $x_i$ denote the static node features of intersection $i$ (e.g., positional encoding) and let $c_t$ encode the vehicle context at time $t$, we form node embeddings $X = [x_1; \ldots; x_{|V|}] \in \mathbb{R}^{|V| \times d}$ and a context embedding $c_t \in \mathbb{R}^d$, then compute:

$$Q = c_t W_Q \in \mathbb{R}^{1 \times d_h}, \quad K = X W_K \in \mathbb{R}^{|V| \times d_h}, \quad V = X W_V \in \mathbb{R}^{|V| \times d_h},$$

and the masked cross-attention weights is obtained by:

$$a = \text{softmax}\left(\frac{QK^\top}{\sqrt{d_h}} + \lambda_r\right) \in \mathbb{R}^{1 \times |V|}, \tag{3}$$

where $\lambda_r(1, i) = -\infty$ prohibits intersections outside the communication radius, and $0$ otherwise. We define the utility vector by the attention scores, i.e., $\theta = a W_\theta \in \mathbb{R}^{|V|}$. This design yields utility scores $\theta$ that (i) respect communication constraints, (ii) capture long-range relevance through attention, and (iii) remain fully differentiable for end-to-end training.

## 4.2 ESTIMATING TRAVEL COSTS GIVEN PARTIAL OBSERVABILITY

At each decision step $t$, multiple vehicles $\mathcal{N}_t$ are simultaneously active in the network. Each vehicle $n \in \mathcal{N}_t$ and selects a subset $S_t^{(n)} \subseteq V$ for V2I communication based on the context-aware utility estimation as introduced above. For each queried pair $(n, s)$ with $s \in S_t^{(n)}$, the traffic signal at intersection $s$ returns a local observation $o_{t,s}^{(n)} \in \mathbb{R}^{d_{\text{obs}}}$ (i.e., signal phase encoding, queue length at incoming and outgoing edges, average speed at incoming and outgoing edges).

At each decision step $t$, every active vehicle $n \in \mathcal{N}_t$ obtains its own partial observation set $O_t^{(n)}$ along with the mask $M_t^{(n)}$ from queried intersections. We treat these partial views as independent

samples in a batched forward pass of a spatio-temporal encoder, producing vehicle-specific global state estimation $\hat{X}_t^{(n)}$. This batched formulation exploits parameter sharing to handle all vehicles simultaneously, substantially reducing the overhead compared to executing distinct networks for each vehicle. Each $\hat{X}_t^{(n)}$ provides a dense estimate of edge-level travel costs tailored to the querying vehicle's perspective.

To obtain global traffic dynamics $\hat{X}_t$ from such sparse and irregular inputs, we employ a spatio-temporal aware transformer encoder (Yun et al., 2019; Dwivedi & Bresson, 2021). Conventional graph convolutional networks are limited by their local message-passing nature, requiring many layers to capture long-range dependencies and thus suffering from over-smoothing (Li et al., 2018a; Oono & Suzuki, 2020). Similarly, recurrent sequence models are biased toward short-term correlations and struggle to represent non-stationary traffic patterns such as periodic rush-hour peaks or abrupt incidents (Bai et al., 2018; Vaswani et al., 2017). In contrast, the transformer encoder integrates structural priors with attention-based aggregation, enabling intersections to attend selectively to both nearby and distant nodes while respecting road connectivity via adjacency-derived masks (Veličković et al., 2018). Temporal context is incorporated directly into the attention computation (Zhou et al., 2021; Wu et al., 2020), allowing the model to align present observations with relevant historical states. By stacking $L$ such layers, we obtain latent embeddings $H_t^{(n)} \in \mathbb{R}^{|V| \times d_{\mathrm{st}}}$ that summarize observed signals, structural dependencies, and temporal dynamics for each vehicle $n$. A lightweight decoder $g_\psi$ then maps these embeddings into dense edge-level travel cost estimates, yielding vehicle-conditioned predictions of link travel costs $\hat{w}^{(n)}(t)$. These estimates feed directly into the differentiable routing head, ensuring that the encoder is trained not merely for predictive accuracy but for decision-aligned performance in downstream re-routing.

### 4.3 Differentiable Routing Decision-Making

Given the estimated, time-varying edge costs $\hat{w}(t) \in \mathbb{R}^{|E|}$, we seek a decision layer that maps these costs into routing actions while remaining compatible with gradient-based training. The key challenge is to replace the discrete shortest-path $\arg\min$ with a smooth, causality-aware operator, so that gradients can propagate back to upstream modules such as V2I acquisition and full state inference. This design follows the broader principle of differentiable structured optimization, where discrete combinatorial decisions are relaxed into continuous surrogates to align predictive learning with downstream task objectives.

**Time-expanded soft shortest path.** A central challenge in dynamic re-routing is that path costs can only be evaluated sequentially over time: the total travel time of a path depends on both edge weights and their temporal evolution. Classical dynamic programming formulations address this via Bellman recursions on a time–expanded graph, but the use of hard $\min$ operators renders the value function non-differentiable and thus incompatible with gradient-based training (Amos & Kolter, 2017; Bast et al., 2016a). To overcome this limitation, we adopt a smooth relaxation of the Bellman operator that retains structural and causal constraints while enabling end-to-end learning.

Specifically, we build a time–expanded graph with node snippets $(v, t)$ for $t = 0, \dots, T$, and define a temperature-smoothed value function $V_\tau(v, t)$ by the soft Bellman recurrence

$$V_\tau(v, t) = -\tau \log \sum_{u \in \mathcal{N}^+(v)} \exp\Big( -\tfrac{\hat{w}_{(v,u)}(t) + V_\tau(u, t+\Delta)}{\tau} \Big), \quad t = 0, \dots, T - \Delta,$$

with terminal boundary conditions

$$V_\tau(d, t) = 0, \ \forall t, \qquad V_\tau(v \neq d, T) = +\infty.$$

Here $\mathcal{N}^+(v)$ denotes out-neighbors, and masking ensures that attention respects road topology and causal ordering. As $\tau \to 0$, this formulation converges to the classical shortest-path value; for $\tau > 0$, it yields a smooth fixed-point operator that admits stable differentiation, closely related to the maximum entropy control and linearly solvable MDP frameworks (Ziebart et al., 2008; 2010; Todorov, 2007).

**Policy and soft costs.** The induced maximum-entropy routing distribution at $(v, t)$ is

$$\pi_\tau\big((v \to u) \mid v, t\big) = \frac{\exp\big( -(\hat{w}_{(v,u)}(t) + V_\tau(u, t+\Delta))/\tau \big)}{\sum_{u' \in \mathcal{N}^+(v)} \exp\big( -(\hat{w}_{(v,u')}(t) + V_\tau(u', t+\Delta))/\tau \big)}.$$

This distribution defines a differentiable surrogate for next-hop choices and a soft path cost for an origin $(s, t_0)$:

$$\mathcal{L}_{\text{route}} := V_\tau(s, t_0) \; + \; \gamma \, \mathcal{H}\big(\{\pi_\tau(\cdot \mid v, t)\}\big),$$

where $\mathcal{H}$ is an optional entropy regularizer and $\gamma \geq 0$ balances exploration and stability.

**Gradient propagation.** Since $V_\tau$ is the fixed point of a smooth Bellman map, we backpropagate through the layer by implicit differentiation of the fixed-point equation, yielding stable gradients $\nabla_{\hat{w}} \mathcal{L}_{\text{route}}$ that flow into the routing head and further upstream into the spatio-temporal transformer and the utility learning module. This mirrors the principle of turning discrete structured decisions into differentiable surrogates so that training aligns predictors with downstream decision objectives.

With such design, the routing head provides a principled, differentiable surrogate to dynamic shortest-path optimization: (i) the structural–causal mask used in the Bellman updates matches the topology and causality masks in the encoder, ensuring consistency across modules; (ii) the temperature $\tau$ serves as a bias–variance knob, where smaller $\tau$ produces sharper, near-discrete paths while larger $\tau$ yields smoother gradients; (iii) the computational complexity scales with the size of the time-expanded frontier and the sparsity of $\mathcal{N}^+(v)$, thereby preserving scalability on large urban networks; and (iv) most importantly, the entire module is decision-aligned, enabling gradients from downstream routing performance to flow backward into state inference and query selection. This integration ensures that proactive observation acquisition, global travel cost inference, and routing decisions are jointly optimized under a unified end-to-end learning objective.

## 5 EXPERIMENTS

### 5.1 DATASETS AND SIMULATION

We evaluate on two city-scale networks widely used in urban traffic control studies: **Jinan** and **Hangzhou** (Wei et al., 2019; Sun et al., 2023; Zhou et al., 2024), with 12 and 16 intersections. Each network is imported into the SUMO simulator (Krajzewicz et al., 2002) with network geometry, default signal plans, and realistic speed limits. We generate vehicular flows at the start of the simulation with 50 vehicles, randomly assigning each vehicle with an Origin–Destination (OD) pair to simulate real-world traffic. During the simulation, once a certain vehicle completes its trip, a new vehicle is simulated to enter the network. The decision steps are $\Delta_{\text{plan}} = 10s$. At each decision step, $k = 4$ intersections are to be selected via V2I. For both networks, the communication radius is restricted to $2L$ where $L$ is the maximum edge length over the road network.

### 5.2 TRAINING DETAILS

The model is trained end-to-end by minimizing the differentiable routing loss described previously. Specifically, the soft Bellman operator provides a smooth surrogate of the remaining travel time $V_\tau(v, t)$ for each active vehicle, and the route loss, i.e.,

$$\mathcal{L}_{\text{route}}(t_0) = \frac{1}{|\mathcal{A}_{t_0}|} \sum_{n \in \mathcal{A}_{t_0}} V_\tau(s_n, t_0) + \gamma \sum_{n \in \mathcal{A}_{t_0}} \sum_{t \geq t_0} m_{n,t} \, \mathbb{H}\Big[\pi_\tau(\cdot \mid v_t^{(n)}, t)\Big],$$

is computed at each decision step, where $\mathcal{A}_{t_0}$ denotes active vehicles, $s_n$ their current nodes, $m_{n,t}$ is a mask indicating whether vehicle $n$ is still active at time $t$, and $\gamma = 0.01$ balances the entropy regularizer. Gradients are backpropagated through the routing head by implicit differentiation of the soft Bellman fixed point, ensuring updates propagate to both the spatio-temporal encoder and the V2I selection module.

**Data format.** Each half-hour SUMO episode simulation generates a sequence of decision steps, at which vehicles actively query intersections via V2I. At each step $t$, an active vehicle $n$ produces:

- **Partial observation** $O_t^{(n)} = \{o_{t,s}^{(n)}\}_{s \in S_t^{(n)}}$, containing queried local features including queue lengths, approach speeds, and signal phases;

- **Visibility mask** $M_t^{(n)} \in \{0, 1\}^{|V|}$, marking which intersections are observed at time $t$;

- **Vehicle context**, including its current node $v_t^{(n)}$ and destination $d^{(n)}$, as well as corresponding approaching directions.

- **Estimated travel cost**, the predicted edge travel costs $\hat{w}^{(n)}(t)$, from which the routing head computes the policy $\pi_\tau(\cdot \mid v_t^{(n)}, t)$ and value $V_\tau(s_n, t)$ used in the loss.

It is important to note that our framework does not rely on ground-truth travel cost labels. Unlike conventional predict-then-optimize pipelines that train predictors against MSE targets, our supervision comes directly from the decision objective: the soft Bellman operator defines a differentiable surrogate of the remaining travel time, which naturally serves as the training signal. This design eliminates the need for dense cost annotations, which are often unavailable in real deployments, and aligns learning with the ultimate goal of minimizing travel time.

**Hyperparameters.** The spatio-temporal transformer is configured with $L = 2$ encoder layers, hidden dimension $d_{\text{model}} = 64$, and $H = 4$ attention heads. The temperature parameter $\tau$ in the soft Bellman operator is annealed from $0.8$ to $0.2$ over training epochs to gradually sharpen the value function approximation. The entropy regularization coefficient is fixed to $\gamma = 0.01$. The Adam optimizer is used with learning rate $2 \times 10^{-4}$. The model is trained for $100$ epochs. All reported results are averaged over 5 random seeds.

## 5.3 Comparable Methods

We compare our end-to-end approach against *select-predict-navigate* baselines as described below, which first select intersections, then predict edge-level travel costs, and then solve a shortest-path problem with a non-differentiable solver.

**Selection strategies.** We implement two heuristic strategies for V2I query selection: (1) *Random-k*, where $k$ intersections are sampled uniformly at random; (2) *Proximity-k*, selecting the $k$ intersections closest to the current vehicle location.

**Prediction models.** Given selected observations, we compare several representative spatio-temporal predictors trained with MSE loss: (1) *STGCN* (Yu et al., 2018), a spatio-temporal convolutional network; (2) *DCRNN* (Li et al., 2018b), a diffusion convolutional recurrent network. For these baselines, ground-truth supervision is obtained from SUMO by querying edge-level travel times via the TraCI API *(traci.edge.getTraveltime(edgeID))*, which provides per-step labels for training predictive models. These predictors estimate edge-level travel times that are subsequently fed into a routing solver.

**Optimization solvers.** We adopt classical shortest-path solvers as the optimization back-end to compute vehicle routes from predicted edge costs:

(1) *Dijkstra's* (Dijkstra, 1959), a label-setting method that incrementally expands the frontier of nodes in order of increasing tentative distance from the source. (2) *A\* search* (Hart et al., 1968), generalizes Dijkstra by incorporating a heuristic function $h(v)$ that estimates the remaining cost from node $v$ to the target.

**Decision-aware methods.** For a stronger PtO baseline, we additionally include the Smart Predict-then-Optimize (SPO+) baseline (Elmachtoub & Grigas, 2022). Specifically, SPO+ defines a convex surrogate that upper-bounds the routing regret $c^\top y(\hat{c}) - c^\top y(c)$, and whose subgradient with respect to $\hat{c}$ depends only on these optimal solutions. In our implementation, the routing solver A\* remains fully discrete and is used as a black-box oracle inside the SPO+ loss, while gradients are backpropagated through the prediction model.

Together, these components yield a spectrum of nine baselines that serve as natural comparison methods for our end-to-end framework.

Table 1: Main results on Jinan and Hangzhou Networks. AvgTT = Average Travel Time, Delay = Average Delay, AvgWT = Average Waiting Time, all in seconds.

| Method | Jinan | | | Hangzhou | | |
|---|---|---|---|---|---|---|
| | AvgTT↓ | Delay↓ | AvgWT↓ | AvgTT↓ | Delay↓ | AvgWT↓ |
| Random-STGCN-Dijkstra | 191.41 | 57.83 | 39.65 | 290.70 | 79.56 | 56.00 |
| Random-STGCN-A* | 191.41 | 57.83 | 39.65 | 290.70 | 79.56 | 56.00 |
| Random-DCRNN-Dijkstra | 189.56 | 56.43 | 38.27 | 286.31 | 77.02 | 54.32 |
| Random-DCRNN-A* | 189.56 | 56.43 | 38.27 | 286.31 | 77.02 | 54.32 |
| Proximity-STGCN-Dijkstra | 184.29 | 55.01 | 38.11 | 283.09 | 73.99 | 51.34 |
| Proximity-STGCN-A* | 184.29 | 55.01 | 38.11 | 283.09 | 73.99 | 51.34 |
| Proximity-DCRNN-Dijkstra | 185.71 | 55.78 | 38.19 | 283.81 | 74.52 | 51.89 |
| Proximity-DCRNN-A* | 185.71 | 55.78 | 38.19 | 283.81 | 74.52 | 51.89 |
| NN-STTransformer-SPO | 181.72 | 54.09 | 37.00 | 282.41 | 73.78 | 50.82 |
| **E2E (ours)** | **180.96** | **53.41** | **36.74** | **281.79** | **73.11** | **50.20** |

## 5.4 EXPERIMENTAL RESULTS

**Overall performance.** Across both cities and all metrics, the end-to-end (E2E) framework achieves consistent significant improvements. On **Jinan**, E2E reduces *AvgTT*, *Delay*, and *AvgWT* by **1.81%**, **2.91%**, and **3.59%**, respectively, over the strongest PtO baseline (Proximity–STGCN). On **Hangzhou**, the corresponding relative gains are **0.46%** (*AvgTT*), **1.19%** (*Delay*), and **2.22%** (*AvgWT*). These gains are obtained under the same per-step query budget and horizon, indicating that improvements stem from *how* information is acquired and used, rather than from more information. The decomposition of baselines makes clear which design choices matter: (i) *Selection* dominates *predictor* and *solver*. Moving from Random to Proximity selection already improves the PtO pipeline (e.g., for *Jinan*, STGCN: 191.41→184.29, **3.72%** better in terms of *AvgTT*; for *Hangzhou*, STGCN: 290.70→283.09, **2.62%** improvements). This confirms that *which* intersections are queried matters substantially under partial observability. Our E2E model then *further* improves over the best heuristic selection (e.g., *Jinan AvgWT*: 38.11 improves **3.59%**), validating the advantage of **learning** the selection policy jointly with downstream objectives. (ii) *Predictor choice* (STGCN vs. DCRNN) has a secondary effect relative to selection. Under Random selection, DCRNN is slightly better than STGCN on Jinan (189.56 vs. 191.41 in terms of *AvgTT*), but with Proximity selection STGCN becomes the strongest PtO baseline (184.29 vs. 185.71). This interaction suggests that our model is robust against different choices of spatial-temporal modules for the predictor. (iii) *Solver choice* (Dijkstra vs. A*) does not change outcomes—numbers are identical by construction—because both compute the same optimal path under the same edge costs.[1] This controls away confounds in the routing layer and isolates the effect of *information acquisition* and *cost inference*. (iv) Finally, for the NN-STTransformer-SPO baseline, on Jinan, it improves over Proximity–STGCN by about **1.4%**, **1.7%**, and **2.9%** in terms of *AvgTT*, *Delay*, and *AvgWT*, respectively, while on Hangzhou it brings additional gains of roughly **0.2%** (*AvgTT*), **0.3%** (*Delay*), and **1.0%** (*AvgWT*). However, E2E still yields a further **0.4%** (*AvgTT*), **1.3%** (*Delay*), and **0.7%** (*AvgWT*) improvement over NN-STTransformer-SPO on Jinan, and about **0.2%** (*AvgTT*), **0.9%** (*Delay*), and **1.2%** (*AvgWT*) on Hangzhou. The above experimental results indicate that SPO-based PtO already constitutes a strong decision-aware baseline that substantially improves over classical PtO, yet our proposed decision-focused pipeline still provides additional gains.

**Further Experiments.** Due to space limitations, we attach ablation studies and parameter sensitivity analysis of $k$ in the Appendix.

---

[1]With a consistent heuristic, A* expands fewer nodes but returns the *same* shortest path cost as Dijkstra; our metrics depend only on path costs, not on search efficiency.

## 6 CONCLUSION

In this work, we introduced a fully differentiable framework for real-time urban navigation under partial observability. By unifying V2I information acquisition, travel cost inference from partial observations, and differentiable routing optimization, our method directly aligns model training with the downstream objective of minimizing travel time. The learnable selection module adaptively queries the most informative intersections under strict communication constraints, while the spatio-temporal encoder reconstructs network-level travel costs from sparse signals. A differentiable routing layer then computes efficient re-routing strategies and enables end-to-end gradient flow. Experiments on large-scale traffic simulations demonstrated consistent improvements in travel efficiency over various baselines, despite relying on limited communication bandwidth. Beyond advancing algorithmic performance, our framework highlights the importance of jointly modeling sensing, prediction, and decision-making for intelligent transportation systems. We believe this approach provides a scalable pathway toward deployment in collaborative vehicle-infrastructure systems.

## 7 LLM USAGE

The Use of Large Language Models (LLMs) in this paper includes to aid or polish writing, and for paper retrieval and discovery such as finding related work.

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

## A   ABLATION STUDIES

We conduct ablation studies on both city networks to quantify the contribution of each component of our framework, using the following model variants.

**w/o learnable selection (Random).**   This variant replaces the IMLE-based top-$k$ selector with a uniform random strategy that selects $k$ intersections at each decision step. The encoder and differentiable routing layer are kept unchanged and are still trained with the decision-focused loss.

**w/o learnable selection (Proximity).**   Here we use a hand-crafted heuristic that always queries the $k$ spatially closest intersections to the vehicle. This serves as a strong PtO-style heuristic for selection while sharing the same encoder and routing head as the full model.

**w/o vehicle context.**   This variant removes vehicle-specific context. It evaluates the importance of vehicle-level intent for effective V2I querying and routing.

**w/o spatio-temporal reconstruction.**   To study the effectiveness of dense V2I information reconstruction, we replace the spatio-temporal encoder with a single layer MLP.

Table 2: Ablation studies on Jinan and Hangzhou networks. AvgTT = Average Travel Time, Delay = Average Delay, AvgWT = Average Waiting Time, all in seconds.

| Method | Jinan | | | Hangzhou | | |
|---|---|---|---|---|---|---|
| | AvgTT↓ | Delay↓ | AvgWT↓ | AvgTT↓ | Delay↓ | AvgWT↓ |
| Ours | 180.96 | 53.41 | 36.74 | 281.79 | 73.11 | 50.20 |
| w/o learnable selection (Random) | 188.32 | 55.52 | 38.19 | 292.16 | 75.84 | 52.02 |
| w/o learnable selection (Proximity) | 186.39 | 55.14 | 38.15 | 284.61 | 73.43 | 50.70 |
| w/o vehicle context | 210.83 | 63.17 | 43.86 | 311.02 | 81.57 | 55.01 |
| w/o spatio-temporal reconstruction | 200.07 | 58.91 | 40.72 | 311.22 | 80.57 | 55.81 |

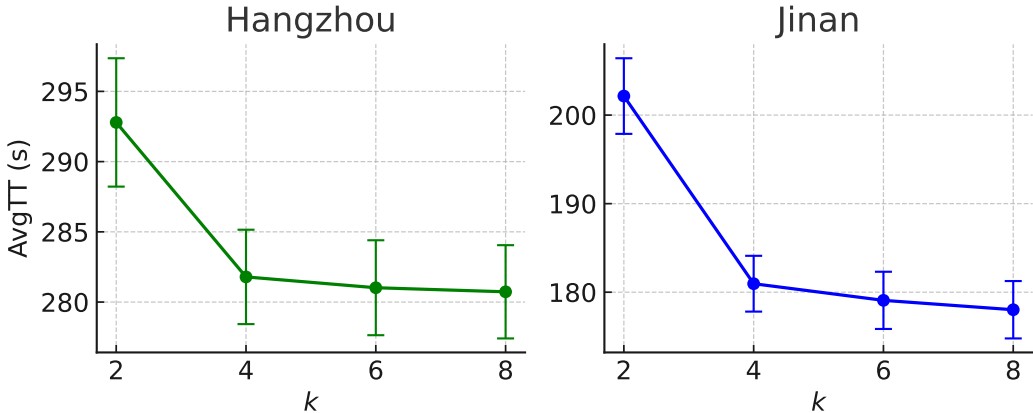

Figure 2: Sensitivity of average travel time to the V2I query budget $k$.

Table 2 shows that our full model consistently achieves the best performance on both city networks. On Jinan, replacing the learned selector with Random-$k$ increases AvgTT from 180.96 s to 188.32 s (about $+4.1\%$), while Proximity-$k$ leads to a smaller but still noticeable increase to 186.39 s (about $+3.0\%$). A similar trend is observed on Hangzhou, where Random-$k$ and Proximity-$k$ increase AvgTT by roughly 3.7% and 1.0%, respectively. These results indicate that simple heuristics already provide competitive query locations, but the IMLE-based selector still yields consistent improvements (about 1–4% across metrics), confirming the benefit of learning V2I acquisition rather than fixing it a priori.

Additionally, removing vehicle context or spatio-temporal reconstruction leads to substantially larger performance degradations. On Jinan, dropping vehicle context increases AvgTT, Delay, and AvgWT by approximately 16.5%, 18.3%, and 19.4%, respectively, while removing spatio-temporal reconstruction yields around 10–11% degradation across all three metrics. The pattern is consistent on Hangzhou: AvgTT worsens by about 10.4% for both ablations, and Delay/AvgWT also degrade by roughly 10–12%. Overall, these results highlight that (i) vehicle-level intent and local context are crucial for choosing informative V2I queries and making effective routing decisions, and (ii) reconstructing dense spatio-temporal information from sparse queries is a key driver of performance, beyond what can be achieved by a shallow MLP over raw observations.

## B  PARAMETER SENSITIVITY ANALYSIS

We further examine the sensitivity of our method to the per-step V2I query budget $k$ by varying $k \in \{2, 4, 6, 8\}$ and reporting the resulting AvgTT metric, as shown in Figure 2.

On the Hangzhou network, increasing $k$ from 2 to 4 reduces AvgTT from $292.79 \pm 4.57$ s to $281.79 \pm 3.36$ s (about 3.8% improvement), while further enlarging the budget to $k = 6$ and $k = 8$ only yields marginal gains ($281.02 \pm 3.38$ s and $280.73 \pm 3.32$ s, respectively). A similar trend is observed on Jinan: AvgTT drops markedly from $202.15 \pm 4.27$ s at $k = 2$ to $180.96 \pm 3.15$ s at $k = 4$ (around 10.5%), whereas increasing $k$ to 6 and 8 gives smaller improvements to $179.08 \pm 3.23$ s and $178.02 \pm 3.24$ s. Overall, these results indicate that the proposed framework is not overly sensitive to the precise choice of $k$ once a moderate budget is available: most of the performance gain is already achieved at $k = 4$, with only diminishing returns beyond this point. This justifies our choice of $k = 4$ in the main experiments as a reasonable trade-off between communication cost and routing performance.

## C  Pseudocode

---

**Algorithm 1: End-to-end training for proactive V2I selection with a routing decision–informed soft Bellman loss**

---

**Input:** Directed graph $G = (V, E)$; adjacency $A \in \{0,1\}^{|V| \times |V|}$; planning step $\Delta$; horizon $T$; OD set $\mathcal{R}_T = \{(s_n, d_n, t_n^{\text{dep}})\}_{n=1}^N$; Top-$k$ budget $k$; Simulator Env with V2I query Query$(\cdot)$ and step Step$(\Delta)$.

**Output:** Scoring network parameters $\theta$; spatio-temporal encoder parameters $\psi$.

1 **Hyperparameters:** soft-Bellman temperature $\tau > 0$; entropy weight $\gamma = 0.01$; history length $T_{\text{hist}}$.

2 **Buffer:** $X_{\text{hist}} \in \mathbb{R}^{T_{\text{hist}} \times |E| \times d}$, $M_{\text{hist}} \in \{0,1\}^{T_{\text{hist}} \times |V|}$.

3 **Params init:** selector network $h_\phi$ (utilities $\theta$), spatio-temporal encoder $g_\psi$ (edge costs $\hat{w}$).

4 **for** *each episode* **do**

5     Env.Reset$(\mathcal{R}_T)$; $t \leftarrow 0$

6     **while** $t \leq T - \Delta$ **do**

7        Define active vehicle set $\mathcal{A}_t$; $C_t \leftarrow \{c_t^{(n)}\}_{n \in \mathcal{A}_t} \in \mathbb{R}^{|\mathcal{A}_t| \times d^{\text{ctx}}}$; static node encoding

         $X \in \mathbb{R}^{|V| \times d^{\text{node}}}$; feasibility mask $\lambda_{\text{r}} \in \{-\infty, 0\}^{|\mathcal{A}_t| \times |V|}$.

       `// Context-aware utility estimation via cross-attention`
       `   between vehicle contexts and node features.`

8        $K \leftarrow X W_K \in \mathbb{R}^{|V| \times d_h}$;   $V \leftarrow X W_V \in \mathbb{R}^{|V| \times d_h}$;   $Q \leftarrow C_t W_Q \in \mathbb{R}^{|\mathcal{A}_t| \times d_h}$;

9        $a \leftarrow \text{softmax}\left(\frac{Q K^\top}{\sqrt{d_h}} + \lambda_{\text{r}}\right) \in \mathbb{R}^{|\mathcal{A}_t| \times |V|}$;   $\theta \leftarrow a W_\theta \in \mathbb{R}^{|\mathcal{A}_t| \times |V|}$.

       `// V2I selection by perturb-and-MAP (shared noise) under a`
       `   strict hard-k budget.`

10       Sample $\epsilon \sim \text{Gumbel}(0,1)^{|\mathcal{A}_t| \times |V|}$;

11       $S_t \leftarrow \text{TopK}(\theta, k) \in \mathbb{N}^{|\mathcal{A}_t| \times k}$;   $M_t \leftarrow \left(\mathbb{I}[v \in S_t^{(n)}]\right)_{n \in \mathcal{A}_t, v \in V} \in \{0,1\}^{|\mathcal{A}_t| \times |V|}$;

12       $\tilde{S}_t \leftarrow \text{TopK}(\theta + \epsilon, k) \in \mathbb{N}^{|\mathcal{A}_t| \times k}$;   $\tilde{M}_t \leftarrow \left(\mathbb{I}[v \in \tilde{S}_t^{(n)}]\right)_{n \in \mathcal{A}_t, v \in V} \in \{0,1\}^{|\mathcal{A}_t| \times |V|}$;

13       $O_t \leftarrow \left\{\{o_{t,s}^{(n)}\}_{s \in S_t^{(n)}}\right\}_{n \in \mathcal{A}_t}$;     `// Context-aware V2I views at time t`

       `// Batched spatio-temporal encoding of sparse and`
       `   irregular observations.`

14       $H_t \leftarrow \text{ST-Encoder}_\psi\left(X_{\text{hist}}, M_{\text{hist}}, O_t, M_t, A\right)$;

       `// Decode full edge-level travel cost estimates.`

15       $\hat{W}_t \leftarrow g_\psi(H_t) \in \mathbb{R}^{|E| \times 1}$;

       `// Soft dynamic shortest-path on a time-expanded graph.`

16       **foreach** $n \in \mathcal{A}_t$ **do**

17         $V_\tau^{(n)}, \pi_\tau^{(n)} \leftarrow \text{SoftBellmanRouting}(\hat{w}^{(n)}(\cdot), A, \tau)$;

         $a_t^{(n)} \leftarrow \text{SampleOrGreedy}(\pi_\tau^{(n)}(\cdot \mid s_t^{(n)}, t))$;     `// Next-hop choice at`
         `current node` $s_t^{(n)}$

18       **end**

19       $a_t \leftarrow \{a_t^{(n)}\}_{n \in \mathcal{A}_t}$;

       `// Apply routing actions, advance simulation, and`
       `   accumulate differentiable routing loss.`

20       $(r_t, \text{state}_{t+\Delta}) \leftarrow \text{Env.Step}(a_t)$;

21       $\mathcal{L}_{\text{route}}(t) \leftarrow \frac{1}{|\mathcal{A}_t|} \sum_{n \in \mathcal{A}_t} V_\tau^{(n)}(s_t^{(n)}, t) + \gamma \sum_{n \in \mathcal{A}_t} \sum_{t' \geq t} m_{n,t'} \mathbb{H}\left[\pi_\tau^{(n)}(\cdot \mid v_{t'}^{(n)}, t')\right]$;

22       $\mathcal{L}_{\text{route}} \mathrel{+}= \mathcal{L}_{\text{route}}(t)$;

23       $t \leftarrow t + \Delta$;

24     **end**

     `// Backpropagation.`

25     $\mathcal{L}_{\text{IMLE}} \leftarrow \sum_t \mathcal{L}_{\text{IMLE}}(S_t, \tilde{S}_t)$;

26     $\mathcal{L} \leftarrow \sum_t \mathcal{L}_{\text{route}}(t) + \lambda_{\text{IMLE}} \mathcal{L}_{\text{IMLE}}$;

27     Update parameters $\{W_K, W_Q, W_V, W_\theta, \psi\}$ by one gradient step on $\mathcal{L}$;

28 **end**

---

