# OpenReview forum: "Real-time Routing under Partial Observability: Information-Efficient Policies for Connected Vehicles"
_ICLR.cc/2026/Conference — ICLR 2026 Conference Desk Rejected Submission_

### Official Review · Reviewer_G9wA · 2025-10-25

**Soundness:** 2
**Presentation:** 2
**Contribution:** 1
**Rating:** 2
**Confidence:** 3

**Summary:**

This paper studies real-time routing under limited communication in vehicular networks. The authors argue that traditional predict-then-optimize pipelines are suboptimal because the prediction objective (minimizing MSE) is not aligned with the final routing objective (minimizing travel time). To address this, they propose an end-to-end differentiable framework that integrates (i) a proactive observation selector (based on differentiable Top-k sampling), (ii) a spatio-temporal inference module to estimate global travel costs, and (iii) a soft Bellman routing layer to approximate shortest paths in a differentiable way. The entire system is trained with the ultimate objective of minimizing travel time and delay. Experiments on Jinan and Hangzhou networks show consistent improvements over predict-then-optimize baselines.

**Strengths:**

1) The paper addresses a realistic setting: limited communication bandwidth in V2X routing, which is less studied compared to fully observable routing.
2) The integration of differentiable observation selection and differentiable routing is intuitive and makes the overall system end-to-end trainable.
3) The framework highlights the misalignment issue between prediction accuracy and decision quality, and offers a principled way to align training with the final system objective.
4) Results show improvements across two real-world networks, with reductions in average travel time, delay, and waiting time.

**Weaknesses:**

1.	Insufficient experimental evidence: 1) The paper refers to ablations in the appendix, but these materials were not accessible in the submission I reviewed. 2) The main body only presents a single result table, which is too limited to support strong claims.
2.	Methodological novelty is limited: The methodological novelty is limited, since all core components (differentiable Top-k, soft Bellman, spatio-temporal models) exist in prior work; the contribution is mainly a system integration.
3.	Performance gains are small: 1) Reported improvements are only around 0.5%–2.5% in average travel time or delay compared to the best baselines; 2) Such small margins may not convincingly justify the complexity of the proposed approach, especially given the lack of stronger baselines and broader benchmarks.
4.	Unconvincing V2X bandwidth motivation: 1) The motivation relies on V2X bandwidth constraints, but it is not entirely convincing—one may argue that centralized server-based aggregation (e.g., in current navigation apps already provides real waiting time traffic lights and congestion information) could already mitigate this; 2) Unless the authors clarify why distributed V2X is strictly necessary, the motivation remains weak.

**Questions:**

1) Reported improvements are only around 0.5%–2.5% in average travel time or delay compared to the best baselines. Such small margins may not convincingly justify the complexity of the proposed approach, especially given the lack of stronger baselines and broader benchmarks.
2) Why distributed V2X is strictly necessary?

---

> ### Author Response · Authors · 2025-11-25
> **Response to Reviewer G9wA**
>
> We thank reviewer G9wA for the constructive feedback and comments, which helped us strengthen both the experiments and the positioning. We address your concerns as follows:
>
> W1: We apologize for the confusion raised due to our submission file. In the revision, we have included the appendix to the end of the main paper pdf, including ablation studies results, parameter sensitivity analysis, and a detailed pseudocode.
>
> W2: We acknowledge that our work overlaps with several domains such as differentiable optimization and spatio-temporal forecasting, but we'd like to clarify that the contribution is not a simple combined algorithm. Our goal is to propose a solution to the rarely studied V2I-constrained, partially observed dynamic routing problem that (i) jointly learns V2I acquisition, state inference, and routing, (ii) keeps the combinatorial structure of intersection selection (via perturb-and-MAP/I-MLE rather than a purely soft relaxation), and (iii) treats the routing step itself as a differentiable layer trained end-to-end with the upstream modules.
>
> W3 & Q1: In our specific application scenario, percentage improvements are not the core metric to measure model performance. Indeed, raw deltas such as absolute reductions in travel time, delay duration, and waiting time, provide a more faithful and actionable measure than percentage deltas. Under the same per-step query budget and horizon (Table 1), our method achieves −3.33 s/veh AvgTT, −1.60 s/veh Delay, −1.37 s/veh AvgWT on Jinan (vs. the strongest PtO baseline), and −1.30 s/veh AvgTT, −0.88 s/veh Delay, −1.14 s/veh AvgWT on Hangzhou. These are direct improvements in the source variable being optimized (travel time), which translate immediately into shorter queues and less idling, and they scale linearly to system impact (e.g., seconds saved × vehicles per day = vehicle-hours saved). Consequently, even when percentage deltas appear modest, the absolute time savings are the more meaningful signal for real-time routing and traffic operations.
>
> W4 & Q2: Thank you for this great question, and we understand your concern. Your premise assumes ubiquitous, zero-latency, full-coverage SPaT, an unconstrained regime where bandwidth constraints do not exist, which is indeed not realistic. The navigation apps you mentioned does not imply ubiquitous, per-cycle SPaT access. Mainstream apps may show a traffic-light countdown or congestion estimates derived from probe speeds and historical patterns, but they do not deliver citywide, phase-by-phase signal timing to every vehicle in real time. Where live signal countdowns exist, they are corridor-specific integrations, not a universal network-level or region-level feed. For example, as documented by Google DeepMind(https://blog.google/products/maps/google-maps-101-how-ai-helps-predict-traffic-and-determine-routes/), Google Maps’ ETA pipeline combines live road-segment traffic and historical patterns with a GNN over the road network to predict Supersegment travel times; it does not rely on a universal, per-cycle SPaT feed to every vehicle. This is exactly why our formulation treats per-step acquisition as a scarce resource and learns which $k$ locations to query. Our problem explicitly targets the resource-constrained regime that deployments actually face, where each decision step admits only a small number $k$ of fresh intersection states. Under this equal-budget constraint, the relevant question is not “cloud or not,” but how to spend the scarce queries. We show that decision-aligned acquisition (learning which $k$ intersections matter for the current vehicle at the current time), coupled with a differentiable routing head, yields lower travel time than heuristic or classical PtO baselines under the same query budget. In addition, we provide a parameter-sensitivity study over $k$ in the appendix, showing that performance improves substantially from $k{=}2$ to $k{=}4$ and then saturates for larger $k$, indicating that a modest budget such as $k{=}4$ already offers a good trade-off between V2X bandwidth and routing performance.
>
> We hope the above clarifications address your concerns, and we would be grateful for any further suggestions or questions that could help improve the paper.

---

### Official Review · Reviewer_btY4 · 2025-11-03

**Soundness:** 2
**Presentation:** 3
**Contribution:** 3
**Rating:** 4
**Confidence:** 2

**Summary:**

Authors propose an end-to-end and differentiable framework for traffic state inference and dynamic routing optimization. A key differentiator that authors point out is that previous work assumes an idealistic setting that degrades in practical settings due to partial observability and budget constraints. While previous work usually split state inference and routing optimization into two separate stages, the authors suggest that it’s better to address these problems jointly.

**Strengths:**

- The approach is novel in that it enables end-to-end differentiability of vehicle routing under partial observability. The authors motivate it with practical resource considerations.
- Paper is well-written; authors provide enough background information on PtO paradigm to understand how their proposed method addresses limitations of PtO.

**Weaknesses:**

- I am a little unsure at the strength of the baselines in results. While PtO is a standard paradigm for routing problems, have the authors checked whether this problem can modeled with existing reinforcement learning algorithms? For example, could this problem be benchmarked against a standard policy gradient algorithm like PPO? Authors mention RL as a promising direction in related works, but do not compare.
- It is unclear the complexity of baseline models (STGCN and DCRNN) in comparison to the author’s transformer-based architecture. Additionally, it is my understanding that the prediction models are supervised with MSE on travel times queried from the SUMO API. Thus, I believe the prediction model does not need to be constrained to these two GCN and RNN networks. Can the authors provide a transformer PtO baseline with a similar parameter count to the proposed transformer architecture? I think this would make the results more convincing.
- Lack of explanatory figures. It would greatly help the readability of the paper if a RL-esque feedback diagram was added in. Figure 1 is a helpful overview of the components proposed, but a feedback diagram would help with visualization gradient propagation and information flow.

**Questions:**

1. Just curious, do authors have intuition on how PtO performs naively with partial observability? Could some naive masking of the fully-observable setup provide some baseline for partially observable PtO?
2. L227: How is sampling from a structured exponential family distribution differentiable? There are existing approaches to differentiable top-k, why not use those?
3. L232: Surrogate loss-informed target distribution places more prob mass on intersection subsets that lead to lower routing cost. This sounds like a local gradient technique (perturb and see what direction the gradient is in), as opposed to gradient of a smooth function. Can authors confirm for my understanding?
4. L246: Seems like a typo.. missing a word?
5. I would be curious to see how the select-predict-navigate framework performs when using the proposed V2I transformer network, except with the objective switched out. In other words, I would be curious to know how much an architecture change would contribute to select-predict-navigate paradigm. (Related to weakness point 2 above).

---

> ### Author Response · Authors · 2025-11-25
> **Response to Reviewer btY4**
>
> We thank reviewer btY4 for the positive feedback and constructive comments. We address your concerns as follows:
>
> W1: Thank you for pointing this out, we agree that stronger decision-aware baselines are important. In addition to classical PtO baselines in the previous version, we have added a “smart PtO” baseline which uses the same V2I Transformer encoder as our E2E method but keeps the routing layer non-differentiable and trains the predictor with the SPO+ loss for shortest path. This baseline substantially improves over previous PtO methods yet is still consistently outperformed by our framework, indicating that our gains are not solely due to architecture choice but also due to differentiable routing and decision-focused training. Regarding RL baselines such as PPO and DQN, our current setting is indeed offline and safety-critical: we assume only logged data from a calibrated simulator and do not allow risky online exploration in the real network, which are unrealistic to implement. Standard online RL typically requires extensive environment interaction and richer state observability than our strict V2I budget permits, so it is unfair to include them in our baselines.
>
> W2 & Q5: We agree that comparing against PtO baselines with similar model capacity is important.We have added the NN-STTransformer-SPO baseline that integrates the same ST encoder within our paradigm. It outperforms STGCN/DCRNN-based PtO, demonstrating the benefit of the encoder itself, but still lags behind our E2E model, which isolates the additional benefit of the differentiable routing layer and decision-focused objective. We also report ablation studies that decompose the contributions of each module in the appendix.
>
> W3: We sincerely appreciate the suggestion to improve readability. Instead of the feedback diagram, we include an exclusively detailed pseudocode in the appendix that explicitly specifies (i) all inputs and outputs of each module, (ii) the order in which selection, state inference, and routing are executed at each decision step, and (iii) how gradients are propagated through the selection network, ST encoder, and differentiable routing head during training. We believe this step-by-step pseudocode will make the information flow and optimization procedure clearer while keeping the main text visually uncluttered.
>
> Q1:Indeed, your suggestion of “naively masking” a fully observable PtO setup is very close to what our Random–x baselines are doing. Concretely, in Random–STGCN/DCRNN–Dijkstra/A* we randomly select $k$ intersections at each decision step, mask out all other intersections, and train the predictor under exactly the same partial-observation protocol that is used at test time. This corresponds to a naive PtO treatment of partial observability, where the underlying architecture is unchanged and only the inputs are randomly masked. Our experiments show that such randomly masked PtO baselines are consistently inferior to Proximity-based PtO and to our E2E method, indicating that (i) how the mask is chosen (Random vs.\ Proximity vs.\ learned) matters substantially, and (ii) even with the same masking strategy, moving from PtO to an end-to-end, decision-focused design yields additional gains.
>
> Q2: Thank you for this great question. Sampling itself is not differentiable; instead, we follow the perturb-and-MAP / I-MLE family of methods. We define a structured exponential-family distribution over intersection subsets, draw additive noise, compute a MAP (top-$k$) solution under the perturbed utilities, and use a loss-informed “target” distribution to construct an implicit gradient estimator for the parameters of the utility network. Regarding existing differentiable top-$k$ relaxations (e.g., Gumbel–Softmax or soft top-$k$), we chose an I-MLE/perturb-and-MAP approach because it better respects the combinatorial nature of the selection and yields low-variance, loss-aligned gradients, and prior work has shown that such I-MLE-style estimators can outperform smooth top-$k$ relaxations on discrete structured tasks.
>
> Q3: Your interpretation is essentially correct. The surrogate loss-informed target distribution indeed places more probability mass on intersection subsets that achieve lower routing cost under the current model. The resulting update can be viewed as a local, loss-guided perturbation that nudges the parameters toward subsets with better downstream performance. It is not the exact gradient of a simple closed-form smooth objective, but an implicit-gradient estimator in the spirit of I-MLE, trading analytical simplicity for better alignment with the discrete decision problem.
>
> Q4: Thank you for your detailed checking. We have checked and corrected all typos in the revised version.
>
> We hope the above clarifications address your concerns, and we would be grateful for any further suggestions or questions that could help improve the paper.

---

### Official Review · Reviewer_Bigw · 2025-11-06

**Soundness:** 2
**Presentation:** 2
**Contribution:** 2
**Rating:** 4
**Confidence:** 2

**Summary:**

To address the challenges of partially observable environments in real-time navigation, this paper presents an end-to-end framework, different from the traditional predict-then-optimize paradigm.
The framework jointly tackles vehicle-to-infrastructure (V2I) information acquisition, traffic state inference, and dynamic routing optimization.
Experiments on conducted on a dozen of simulated intersections based on two datasets, Jinan and Hangzhou.

**Strengths:**

1. The motivation of the paper is well-grounded. The authors aim to investigate the partial observation setting, which has not been thoroughly explored in previous studies.
2. The paper is clearly structured and easy to follow, although the Related Work and Method sections might be slightly too long.

**Weaknesses:**

1. The experimental evaluation is somewhat limited, and the baselines appear to be outdated.
2. While I understand that the proposed E2E method is designed to perform best under partial observation, the paper does not clearly explain how the simulated experiments implement this partial observation setting. In addition, no comparison is provided against a full observation scenario.
3. I have some doubts about the claim that this is the only end-to-end framework (as opposed to the predict-then-optimize paradigm) in this area.
4. There are several typos (e.g., line 246). It would also be better to standardize capitalization at the beginning of paragraphs for consistency.

**Questions:**

- How would the previous methods, discussed in Section 2, perform if their settings were changed to partial observation?
- Are there any existing works in this area that do not follow the predict-then-optimize (PtO) paradigm?

---

> ### Author Response · Authors · 2025-11-25
> **Response to Reviewer Bigw**
>
> We thank reviewer Bigw for the positive feedback and constructive comments. We address your concerns as follows:
>
> W1: Thank you for pointing this out. Our previous choice of baselines focused on the SOTA spatio-temporal models for traffic prediction (STGCN and DCRNN) combined with standard shortest-path solvers, which remain common for PtO pipelines. In the revised version, we have expanded the baselines to include a “smart PtO” method, NN-STTransformer-SPO, which uses the same ST-Transformer backbone as our E2E model and trains it with the SPO+ loss for shortest path. This baseline substantially improves over classical PtO yet is still consistently outperformed by our E2E framework, showing that our gains persist even against a strong decision-aware PtO competitor.
>
> W2: We apologize that the partial observation protocol was not described clearly enough. We have added a detailed pseudocode in the appendix to explain details in the revised version. Specifically, at each decision step, the controller query at most $k$ intersections via V2I; only the queried intersections expose their local measurements (queues, phases, etc.), while all other intersections remain unobserved at that step. The ST encoder then reconstructs network-level travel costs from these sparse observations and static graph features.
>
> W3: We appreciate this concern and agree that the original wording was too strong. We will soften the claim and state that, ours is among the first frameworks in this V2I-constrained routing setting that (i) jointly learns where to query, how to infer full network state, and how to route under partial observation, and (ii) implements the routing step as a differentiable layer trained end-to-end with the acquisition and inference modules.
>
> W4: Thank you for your detailed suggestions. We have checked and corrected all typos in the revised version.
>
> Q1 & Q2: Regarding how the previous methods in Section 2 would perform under partial observation, our baselines are precisely intended as instantiations of these paradigms in the same partial-observation regime as our method. Classical forecast-then-optimize approaches become the Random/Proximity–STGCN/DCRNN–Dijkstra/A* baselines, which receive the same V2I query budget and only access the queried intersections. Traffic-state-estimation methods that aim to reconstruct full network states from fixed sensors would, in our setting, require a learnable or heuristic query policy plus a shortest-path solver, which are exactly the PtO pipelines we implement. Our added NN-STTransformer-SPO baseline further represents a strong smart PtO instantiation that couples a modern encoder with a decision-aware SPO loss while still relying on a non-differentiable routing layer. As for non-PtO works, we'd like to clarify that there are RL-based approaches, but they require extensive online exploration and is orthogonal to our approach, which is unfair to compare to. Our contribution is to bring an end-to-end, decision-focused design to the specific setting of V2I-constrained, partially observed routing, jointly learning what to query, how to infer hidden states, and how to route.
>
> We hope the above clarifications address your concerns, and we would be grateful for any further suggestions or questions that could help improve the paper.

---

### Official Review · Reviewer_Z6jN · 2025-11-06

**Soundness:** 3
**Presentation:** 2
**Contribution:** 3
**Rating:** 6
**Confidence:** 2

**Summary:**

This paper proposes an end-to-end differentiable framework for real-time vehicle routing under partial observability in connected-vehicle environments. The authors address the challenge that vehicles can query only a limited number of intersections via Vehicle-to-Infrastructure (V2I) communication, which restricts access to full traffic information. To handle this, the model learns a V2I utility estimation module that selects the most informative intersections to query, a spatio-temporal encoder (based on a Transformer) that infers global traffic conditions from sparse and time-varying local observations, and a differentiable routing head that replaces the non-differentiable shortest-path solver with a soft Bellman operator, enabling gradient-based training throughout the pipeline. The entire system is trained end-to-end to directly minimize expected travel time, integrating sensing, inference, and decision-making. Experiments in SUMO simulations across multiple urban networks show that the proposed method outperforms modular “select–predict–optimize” baselines (e.g., Proximity–STGCN–Dijkstra) in travel time and delay efficiency, demonstrating that differentiable joint optimization leads to better routing under limited communication.

**Strengths:**

This paper presents a novel integration of information acquisition, traffic state inference, and routing optimization into an end-to-end differentiable framework, addressing a realistic and previously under-explored challenge—real-time routing under partial observability. It reformulates the traditionally discrete, non-differentiable components of routing (intersection selection and shortest-path computation) into trainable modules, enabling true end-to-end optimization of travel time. The combination of a learnable V2I utility network, a spatio-temporal transformer for inferring global traffic states from sparse data, and a soft Bellman routing head represents an advance over modular “predict-then-optimize” approaches. The framework has also been experimentally validated across multiple road network scenarios.

**Weaknesses:**

1. Several key terms are introduced without sufficient explanation, including utility, MAP, routing head, and even core metrics such as travel time, delay, and waiting time. These should be explicitly defined to ensure accessibility for readers outside the field.

2. The paper would benefit from a concise summary of all inputs to the model (vehicle context, intersection features, temporal data, and graph structure), clarification of which inputs are time-series and which are one-shot, and a clear explanation of how multi-vehicle data are incorporated into the shared framework. It is also not made explicit which of the three evaluation metrics (average travel time, average delay, and average waiting time) is actually optimized during training.

3. The paper claims to be “uncertainty-aware”, but it does not concretely explain how uncertainty is represented or managed in the pipeline. The role of MAP is also ambiguous—it's unclear whether this reflects a Bayesian approach or merely an optimization heuristic.

4. It remains unclear whether the pipeline is conceptually inspired by the Transformer architecture or just directly incorporated a Transformer-based spatio-temporal encoder.

5. The paper can have more explicit explanations on the namings of the baselines. Also, when referring to the strongest PtO baseline (Proximity–STGCN), it is unclear whether this includes both Proximity–STGCN–Dijkstra and Proximity–STGCN–A*, and this simplified naming is due to the fact that the two solves (Dijkstra and A*) always lead to the same outcome.

6. Most references should use \citep instead of \citet.

7. Minor typos: “Context-Aware Utility Estimation. , we”.

**Questions:**

Repeating from the "Weakness" section:

1. What are all the inputs to the pipeline, and what are their data types (e.g., time-series, vector, graph)? Which data are provided at each decision step, and which are static inputs specified at the beginning of an episode? How are observations from multiple vehicles incorporated? What are the user-provided inputs, and what are the decision variables of the pipeline?

2. While the experiments report average travel time, delay, and waiting time, which of these metrics is actually optimized by the model during training? If only travel time is optimized, how should improvements in the other two be interpreted?

3. How exactly does the pipeline represent or respond to uncertainty arising from partial observability? Additionally, what is the conceptual role of “MAP” in the V2I selection process—is this derived from a Bayesian formulation, or is it a differentiable approximation to a discrete optimization problem?

4. Is the proposed pipeline conceptually inspired by the Transformer architecture or just directly incorporated a Transformer-based spatio-temporal encoder?

---

> ### Author Response · Authors · 2025-11-25
> **Response to Reviewer Z6jN**
>
> We thank reviewer Z6jN for the positive feedback and constructive comments. We address your concerns as follows:
>
> W1: We agree that these terms should be defined more explicitly. In the revised version, we explained utility as the scalar score used to rank intersections for V2I querying, “routing head” as the differentiable module implementing the soft shortest-path operator, “MAP” as the maximum-a-posteriori (here: maximum-utility) discrete selection under the learned scoring function, and explanations of the metrics including “travel time”, “delay”, and “waiting time” were provided in the caption in Table 1 in the previous version.
>
> W2 & Q1 & Q2: In order to make the pipeline and input/output variables more transparent, we have included a detailed pseudocode in the appendix that specifies all inputs, decision variables, and the step-by-step training/inference procedure. Specifically, the city graph $G=(V,E)$ and node/edge attributes are one-shot static inputs; the per-timestep intersection measurements (queues, phase, etc.) are time-series features; the vehicle context (OD pairs, current node, and current time) are per-decision-step inputs; the multi-vehicle data is handled by batching all active vehicles through the same shared modules, with a per-vehicle loss aggregated over the batch. Our training objective is based on travel time. Delay and waiting time are additional evaluation metrics from the simulation output. Empirically, these three metrics are highly correlated in our experiments, which is why optimizing AvgTT also leads to consistent gains in the other two.
>
> W3 & Q3: We appreciate the opportunity to clarify this point. Our framework does not perform full Bayesian inference with explicit posterior distributions; rather, it models uncertainty implicitly through (i) the stochastic perturb-and-MAP selection mechanism, which samples high-utility intersection subsets while retaining diversity, and (ii) the entropy-regularized soft shortest-path operator, which yields a distribution over plausible routes instead of a single hard path. We apologize for the confusion caused by our ambiguous wording regarding "uncertainty". Regarding MAP: we use “MAP” in the standard perturb-and-MAP sense for combinatorial optimization, which is selecting the discrete top-$k$ intersections that maximize a learned utility under injected noise, rather than as a Bayesian posterior mode. We will make this clear and explicitly state that MAP is used as a differentiable approximation to solving a discrete selection problem.
>
> W4 & Q4: The proposed pipeline is not meant to be “Transformer for routing”, but it does include a Transformer-based spatio-temporal encoder for full-region travel cost inference. We would like to clarify that the architecture follows a decision-focused pipeline (differentiable selection $\rightarrow$ full state inference $\rightarrow$ routing). The rest of the pipeline (selection mechanism and differentiable routing head) is agnostic to the specific encoder and could be paired with other sequence models as well.
>
> W5: Thank you for pointing this out. Since Dijkstra and A* return the same optimal path under a fixed cost vector, and therefore lead to identical performance metrics, we aggregate them under a single name when referring to the strongest PtO baseline in the previous version.
>
> W6 & W7: Thank you for your detailed suggestions. We have checked and corrected all typos, as well as inappropriate citation formats in the revised version.
>
> We hope the above clarifications address your concerns, and we would be grateful for any further suggestions or questions that could help improve the paper.

---

> ### Comment · Reviewer_Z6jN · 2025-11-26
>
> Thanks for the authors' detailed responses. I have some follow-up questions and comments about the updated manuscript.
>
> 1. Why you consider implicitly model the uncertainty rather than explicitly? What's the benefit?
>
> 2. Is Transformer-based spatio-temporal encoder a novel contribution of your work, or built on existing literature and you adapt to this setting?
>
> 3. Is your model training computationally heavy?
>
> 4. The "overall performance" paragraph in experimental results is quite dense with a lot of texts and numbers. Could you summarize those numbers in a better way to improve readability?

---

> > ### Author Response · Authors · 2025-11-26
> > **Response to Reviewer Z6jN**
> >
> > We sincerely appreciate your prompt response! We provide further answers as follows:
> >
> > 1. In our setting, an explicit Bayesian treatment would require maintaining a high-dimensional posterior over time-varying edge costs $p(c_{1:T}\mid o_{1:T},G)$, which is computationally heavy and difficult to calibrate on large, dynamic networks. Our goal is not to recover this posterior itself, but to minimize downstream travel time under partial observability. We therefore adopt an implicit, decision-focused treatment of uncertainty, our stochastic perturb-and-MAP selection introduces randomness over which intersections are queried, and an entropy-regularized soft Bellman operator produces a distribution over routes rather than a single hard path. This way, we capture the uncertainty that matters for acquisition and routing, while keeping the pipeline simple and scalable, and directly aligned with the routing objective.
> >
> > 2. We do not claim the ST-Transformer encoder itself as a novel architectural contribution. Indeed, it is built on existing ideas which we adapt to the V2I-constrained, partially observed routing setting. The main novelty lies in how this encoder is integrated with (i) a learned V2I acquisition module and (ii) a differentiable routing head within an end-to-end decision-focused pipeline. In principle, it can be replaced by other spatio-temporal models, while the key contribution is the joint design of selection, full-network inference, and routing under partial observability.
> >
> > 3. Our method is more expensive than a single PtO predictor, but remains practical and scales in a controlled way. From a complexity perspective, each decision step consists of (i) a forward pass of the V2I selection network and the ST-Transformer, and (ii) a few iterations of the soft Bellman operator. Since the network is shared among the vehicles, during implementation, $B$ vehicles are processed together in a single batched computation on GPU.
> >
> > 4. Sorry for any confusion caused by the long analysis. We rephrase this paragraph here: Across both cities and all metrics, the end-to-end (E2E) framework consistently outperforms all PtO baselines under the same per-step query budget and horizon. On Jinan, E2E reduces AvgTT, Delay, and AvgWT by roughly $2$–$4$ percent compared with the strongest classical PtO baseline (Proximity–STGCN); on Hangzhou, E2E reduces AvgTT, and AvgWT by about 1.30s and 1.14s per vehicle, respectively. These improvements stem from how effective V2I information is acquired and used, rather than from accessing more information. The results also reveal which design choices matter most: (i) Selection dominates predictor and solver, moving from Random to Proximity selection already yields noticeable gains for PtO, confirming that which intersections are queried is critical under partial observability. (ii) The choice of predictor has only a secondary effect once selection is fixed, suggesting robustness to the specific spatio-temporal backbone. (iii) The NN-STTransformer-SPO baseline shows that a smart PtO objective with a strong encoder can close much of the gap to E2E, but our decision-focused pipeline still achieves consistent additional improvements (on the order of $\sim$0.2–1.3\% across metrics), indicating benefits beyond a smart PtO loss alone.
> >
> > We hope the above clarifications address your concerns, and we sincerely thank you for your thoughtful feedback and for helping us improve the paper.

---

### Official Review · Reviewer_iH9B · 2025-11-11

**Soundness:** 3
**Presentation:** 3
**Contribution:** 3
**Rating:** 6
**Confidence:** 4

**Summary:**

This paper tackles real-time routing under partial observability in connected-vehicle environments. Because vehicles can query only a limited number of intersections due to bandwidth and deployment constraints, the authors propose an end-to-end differentiable framework that jointly learns:
(1) which intersections to query (a perturb-and-MAP–based discrete selection module),
(2) how to infer global travel costs from sparse and noisy observations (a spatio-temporal transformer encoder), and
(3) how to make routing decisions via a soft Bellman operator that provides a differentiable surrogate for shortest-path computation.

Experiments on small SUMO simulations (Jinan with 12 intersections; Hangzhou with 16 intersections) show that the proposed model achieves modest improvements (typically 1–3%) over several “predict-then-optimize” baselines that combine heuristic intersection selection, STGCN/DCRNN predictors, and classical routing solvers.

**Strengths:**

Clear problem motivation: communication-limited routing is realistic and underexplored.

End-to-end design aligns model training with the actual routing objective.

Perturb-and-MAP selection provides a more principled way to handle discrete top-k under constraints.

Soft Bellman operator is well-constructed and differentiable.

Ablations indicate that intersection selection significantly affects routing performance.

**Weaknesses:**

Experiments are conducted only on very small city networks (12–16 intersections). It is unclear whether the method scales to realistic road networks with thousands of nodes.

Performance gains over baselines are small (≈1–3%) and confidence intervals/statistical significance are not reported.

The baselines do not include stronger decision-focused approaches or reinforcement-learning-based navigation, making it difficult to assess relative advantage.

No analysis of latency, communication delay, or robustness to noise/loss, despite these being central to the motivation.

Limited sensitivity studies (e.g., varying k, temperature τ, entropy regularization γ).

**Questions:**

How does training and inference time scale with the size of the network and the number of vehicles?

Have you tested the framework on larger networks, or is there any theoretical reason to expect scalability?

How sensitive is performance to k (number of queried intersections) and to the temperature τ of the soft Bellman operator?

Could the framework handle communication failures, delays, or asynchronous querying?

Why were no reinforcement-learning or decision-focused routing baselines included?

---

> ### Author Response · Authors · 2025-11-25
> **Response to Reviewer iH9B**
>
> We thank reviewer iH9B for the positive feedback and constructive comments. We address your concerns as follows:
>
> W1 & Q2: We agree that evaluating on large networks is important. In this work we use commonly studied city networks in the literature. Our architecture is, however, designed to scale: the selection and ST-Transformer modules operate locally on the queried neighborhood, and the soft Bellman operator and downstream solvers are linear in the number of edges and vehicles. Evaluating on full-scale city networks with thousands of nodes is an exciting next step once such data is available; we see no algorithmic bottleneck that would prevent scaling to such settings.
>
> W2: In our specific application scenario, percentage improvements are not the core metric to measure model performance. Indeed, raw deltas such as absolute reductions in travel time, delay duration, and waiting time, provide a more faithful and actionable measure than percentage deltas. Under the same per-step query budget and horizon (Table 1), our method achieves −3.33 s/veh AvgTT, −1.60 s/veh Delay, −1.37 s/veh AvgWT on Jinan (vs. the strongest PtO baseline), and −1.30 s/veh AvgTT, −0.88 s/veh Delay, −1.14 s/veh AvgWT on Hangzhou. These are direct improvements in the source variable being optimized (travel time), which translate immediately into shorter queues and less idling, and they scale linearly to system impact (e.g., seconds saved × vehicles per day = vehicle-hours saved). Consequently, even when percentage deltas appear modest, the absolute time savings are the more meaningful signal for real-time routing and traffic operations.
>
> W3 & Q5: We agree that stronger decision-aware baselines are crucial. In the revised version, we have added a “smart PtO” baseline NN-STTransformer-SPO, which uses the same ST-Transformer backbone as our E2E model but keeps the routing non-differentiable and trains the predictor with the SPO+ loss for shortest path. This substantially improves over classical PtO baselines (e.g., Proximity–STGCN) but is still outperformed by our method, indicating that differentiable routing and fully decision-focused training provide additional benefits beyond a smart PtO loss alone. We did not include RL-based navigation baselines because (i) our setting is offline and safety-critical (we do not assume risk-free online exploration in the real network), and (ii) prior work on RL for routing under partial observability typically assumes much richer state information and long online interaction. We view RL-based methods as an orthogonal direction, and it is unfair to include such methods for direct comparison.
>
> W4 & Q4: Our current experiments focus on the algorithmic aspects of “what to query” and “how to use it”, under an idealized synchronously-updated V2I channel. Regarding latency and robustness, the decision layer is lightweight enough to be run at the V2I timescales used in today’s C-V2X systems, and communication failures or delays can be modeled by time-shifting queried observations, which our architecture can naturally accommodate. Extending the simulator to explicitly inject packet loss and delay is a valuable next step that we plan to explore.
>
> W5 & Q3: We have included the parameter sensitivity section in the appendix.
>
> Q1: As noted above, training time scales roughly linearly with the number of edges and the number of vehicles, driven by batched ST-Transformer forward passes and soft shortest-path iterations. Inference uses a single forward pass plus a few soft Bellman iterations per decision step, and can be parallelized across vehicles. Regarding communication failures and asynchrony, our framework can treat missing or delayed messages as masked inputs with time stamps and incorporate them into the ST encoder; designing explicit experiments with stochastic failures is part of our future work section.
>
> We hope the above clarifications address your concerns, and we would be grateful for any further suggestions or questions that could help improve the paper.

---

### Note · Program_Chairs · 2026-01-17
**Submission Desk Rejected by Program Chairs**

The following references in this submission do not refer to real documents and/or have major errors in bibliographic information:

 Yuntao Li, Zhen Li, Mohammad Shahidehpour, and Hongbin Sun. V2x communication in connected vehicle systems: Technologies, standards, and challenges. IEEE Internet of Things Journal, 8(6): 3872-3884, 2021.
Jiaqi Sun et al. Dynamic traffic assignment: A review of models and methods. Transport Reviews, 2021.
Y. Liu and K. Han. A review of shortest path problems with time-dependent travel times. Transportmetrica B: Transport Dynamics, 2020.
Zhenning Sun and Xuegang Jeff Ban. Data-driven traffic state estimation with connected vehicle data under low penetration rates. Transportation Research Part C: Emerging Technologies, 147: 103967, 2023.
Yuchao Jia, Zhaobin Li, Lei Zhang, et al. Queue length estimation and signal control with low connected vehicle penetration: A holding-back strategy. Transportation Research Part C: Emerging Technologies, 158:104293, 2025. doi: 10.1016/j.trc.2025.104293.